Optimizing target-to-total DNA ratio in eDNA studies: effects of sampling, preservation, and extraction methods on single-species detection

Andruszkiewicz Allan Elizabeth eallan@uw.edu eaa326@gmail.com 1
Shaffer Megan R. 1
Kelly Ryan P. 1
Parsons Kim 2
1 School of Marine and Environmental Affairs, University of Washington , Seattle , WA , United States of America
2 Conservation Biology Division, Northwest Fisheries Science Center, National Marine Fisheries Service, National Oceanic and Atmospheric Administration , Seattle , WA , United States of America
Franco Bernardo
Electronic publication date: 2025 Oct 30
Publication date: 2025
Volume: 13
Electronic Location ID: e20127
Received 2025 May 20; Accepted 2025 Sep 2
Copyright: ©2025 Andruszkiewicz Allan et al.
Copyright year: 2025
Copyright holder: Andruszkiewicz Allan et al.
License: This is an open access article distributed under the terms of the Creative Commons Attribution License, which permits unrestricted use, distribution, reproduction and adaptation in any medium and for any purpose provided that it is properly attributed. For attribution, the original author(s), title, publication source (PeerJ) and either DOI or URL of the article must be cited.
License URL: https://creativecommons.org/licenses/by/4.0/

Keywords: Environmental DNA, DNA extraction, DNA preservation, Tursiops truncatus, Quantitative PCR, eDNA methods

Funding: The Office of Naval Research N00014-22-1-2719 This material is based upon research supported by the Office of Naval Research under Award Number (N00014-22-1-2719). The funders had no role in study design, data collection and analysis, decision to publish, or preparation of the manuscript.

==============================
There are many decisions to be made when sampling for environmental DNA (eDNA) analysis, whether using a targeted, single-species assay or community-based metabarcoding. Of the entire workflow from sampling water to bioinformatic analyses, the first steps in the process of collecting water, filtering it, and preserving the filter membranes represent major decision points upon which the success of downstream processes depend. Though many previous studies have compared water volume filtered, filter pore size, and preservation and extraction methods, the conclusions are often that they produce different results, but it is unclear which is the optimal approach for a given purpose. Here, rather than provide yet another methods comparison paper, we provide a framework for how to make informed decisions from a methods comparison and, importantly, how to combine data collected via different methodological choices. We investigate (1) the volume of water filtered and the filter pore size and (2) the preservation method and extraction method of samples with a specific lens on how these choices impact the detection of a single targeted species (Atlantic bottlenose dolphin, Tursiops truncatus, via quantitative PCR (qPCR)), although in principle these findings apply to single-species assays more generally. We find that larger pore size filters (5 µm vs. 1 µm) and larger volumes of water (3 L vs. 1 L) maximize the ratio of amplifiable target DNA to total DNA without compromising the absolute detection of target. We also find that maximizing total DNA yield during extraction (phenol chloroform vs. two commercial kits) does not always increase target detection likely due to the concentration of inhibitors and co-extraction of off-target DNA. We also comment on variation including technical and biological variability between replicates, finding that by homogenizing source water before filtering removes much of the biological variation. Finally, we present a statistical model that allows for inclusion of data from samples collected and processed in different ways, enabling researchers to change protocols or include data from other field sampling efforts, thereby opening up more possibilities to extend datasets and analyses.

Introduction

Since the first applications of detecting environmental DNA (eDNA) in water samples, many papers have been written comparing methodological choices in eDNA sample processing. Most of these have used more general primers to look at many species using a metabarcoding approach (e.g., Deiner et al., 2015; Djurhuus et al., 2017; Majaneva et al., 2018; Li et al., 2018; Deiner et al., 2018; Coutant et al., 2021; Bizzozzero et al., 2024; Bowen et al., 2024; Liu et al., 2024). The results of such methods comparisons often demonstrate small differences between either richness or relative abundance of taxa detected but it is unclear which is the “right” or “best” protocol. Other papers have investigated the effect of methodological choices on a single target species (e.g., Liang & Keeley, 2013; Eichmiller, Miller & Sorensen, 2016; Minamoto et al., 2016; Spens et al., 2017; Hinlo et al., 2017; Capo et al., 2020; Mauvisseau et al., 2021; Fukuzawa et al., 2023; García et al., 2024) with some common findings, such as sample volume increases probability of detection, but often reveal conflicting results on the “best” preservation or extraction method.

It is important to clarify what metric is being used to optimize methodological choices. In a multi-species, metabarcoding approach, it is often species richness—although the interpretation of richness observations is complicated by the fact that metabarcoding data are compositional, and so the probability of detecting one species in a mixture depends strongly upon the other species present in that mixture. For a single species assay, researchers often seek to maximize the probability of detecting the species of interest and thus aim to maximize the recovery of eDNA of the species of interest (i.e., “target DNA”). However, total DNA is much easier and cheaper to quantify than target DNA and therefore is sometimes used as a proxy, becoming the metric of optimization, with the underlying assumption that more total DNA will yield more target DNA.

However, the concentration of total DNA is very often not a good proxy for target DNA. For example, smaller pore size filters (e.g., 0.22 or 0.45 µm) are often used when targeting microbes as microbial DNA is abundant relative to metazoan DNA in the environment due to higher cell densities and smaller particle sizes (Power et al., 2023). In contrast, metazoan (including vertebrate) DNA almost necessarily occurs as larger particle sizes from sources like larger animal cells or tissue fragments in the water column. Therefore, larger pore size filters (e.g., 5 µm) more effectively capture metazoan DNA by reducing capture of microbial DNA. Different target taxa accordingly have different optimal protocols.

Furthermore, recovering more total DNA might not always result in the maximum target DNA. Target DNA will always be a small fraction of total DNA, and the rate at which total DNA increases might not be the same as the rate at which target DNA increases. In cases where the target is particularly rare, collecting more total DNA can be akin to creating a proportionally larger haystack in which to find a needle (target DNA).

Here, we focus on the first stages of sample collection and processing by comparing the volume of water sampled, the filter pore size, preservation method, and extraction method for a single target vertebrate species—bottlenose dolphin (Tursiops truncatus)—in shallow (approximately 20 m), nearshore seawater. We evaluate how methodological choices affect total DNA, target DNA, and the ratio of target:total DNA recovery, and we demonstrate how to use a straightforward linear model to combine samples collected with different protocols. Rather than identifying a single “best” protocol, we present a generalizable framework for decision making for targeted eDNA studies and a framework for responsibly combining data collected using different methodological approaches.

In early proof-of-concept eDNA studies, the volume of water collected ranged from very small volumes (e.g., 15 mL (Foote et al., 2012)) to much larger volumes as it was unclear what was required for this new application and tool (see Takahashi et al., 2023 for review). Studies using metabarcoding approaches revealed a positive correlation between the number of unique species detected and the volume of water filters up to a certain point (i.e., species accumulation curves) (Bessey et al., 2020). In studies employing a single targeted species approach (i.e., quantitative polymerase chain reaction (qPCR) or droplet polymerase chain reaction (dPCR) or digital droplet polymerase chain reaction (ddPCR)), larger volumes of water filtered resulted in more reliable detection of the target species (Liang & Keeley, 2013; Capo et al., 2020). A recent review paper found that ∼40% of studies use 1 L of water (Takahashi et al., 2023), which seems to strike the balance of maximizing probability of detection while working within the bounds of what is practical, but little has been done to quantify or understand the relationship between concentrating more target DNA alongside more non-target (or total) DNA. Additionally, the effects of inhibition, whether by chemical inhibitors alone or from large amounts of non-target DNA, are poorly understood but are expected to increase with increasing volume of water filtered or decreasing filter pore size (Opel, Chung & McCord, 2010; Sidstedt, Rådström & Hedman, 2020).

Though the field of eDNA analysis is relatively new, the concepts are not new and have been used for decades for other applications, primarily for detecting and characterizing microbial community composition from various environments (Ogram, Sayler & Barkay, 1987). This resulted in most methodological choices stemming from the microbial field as well. In particular, the pore size of the filter used to capture DNA for microbial work must be extremely small and therefore in eDNA field sampling, the most common pore size filter still used is 0.45 um (Tsuji et al., 2019). This pore size might work very well for capturing bacteria, but if the target taxa are macroorganisms rather than microbes, the target will be co-captured along with many more off-target microbes and thus make it more difficult to detect as the macroogransims are now a rarer target (Power et al., 2023). Therefore, this may not be the best choice, but it has been used by default for many years in eDNA studies.

Similarly, the optimal DNA extraction methods for microorganisms DNA may be very different than those for macroorganisms. For example, phenol-chloroform-isoamyl extractions are known to maximize total DNA recovery (Ramón-Laca, Wells & Park, 2021), but the total DNA recovered may include non-target taxa (i.e., bacteria) that then overwhelm one or more non-microbial target taxa in the DNA extract. In a recent review of eDNA methodology, Tsuji et al. (2019) found that over 75% of papers used a commercial kit for eDNA extraction. Again, the majority of studies comparing pore size, preservation method, and extraction method compare multi-species detection via metabarcoding, finding slight differences that may or may not be correlated with target taxa (Turner et al., 2014; Deiner et al., 2015; Eichmiller, Miller & Sorensen, 2016; Minamoto et al., 2016; Djurhuus et al., 2017; Li et al., 2018; Kumar, Eble & Gaither, 2019; Mauvisseau et al., 2021; García et al., 2024; Rodriguez et al., 2025). However, it is never abundantly clear which is the “optimal” protocol to maximize species recovery and probability of detection.

Part of what makes it difficult to assess and compare different protocols is underlying variability between samples. Here, we define biological replicates as replicate water samples/filters taken from the same environment and technical replicates as replicate reactions for a given molecular analysis (e.g., qPCR) from the same water sample/filter. True biological variability will result in biological replicates generating slightly different results so without accounting for this, it can be difficult to attribute observed differences to either protocols or biological variability. This is very common in presence/absence metabarcoding results of many species, but it is also true for a single species in a quantitative assay such as quantitative PCR (qPCR) or digital or droplet digital PCR (dPCR or ddPCR). Although biological replicates taken next to each other may give slightly different results due to irreducible sampling variance, often comparisons of methods do not explicitly separate biological variability from methodological variability. A few other studies have looked specifically at how variability (as measured by coefficient of variability) can be impacted by methodology as well (Eichmiller, Miller & Sorensen, 2016; Minamoto et al., 2016; Mauvisseau et al., 2021). Here we isolate technical/biological variability and method variability and also look at homogenized versus non-homogenized water to further investigate how different biological replicates are when source water is well mixed versus grab samples.

Finally, it is very common that researchers are interested in combining results from samples collected or processed in different ways. Whether combining data from different published papers or across different experiments where protocols have evolved or adapted over time, the best way to integrate qPCR data across datasets with differing field or laboratory protocols has been challenging. However, given a set of samples collected from a common pool but subsequently treated differently, it becomes possible to isolate the effects of different methodological choices on a defined outcome (here, target-species eDNA concentration), and derive a straightforward model to “correct” observed concentrations for different methodological choices. This shows precisely how different protocol choices affect eDNA quantification and importantly allows for comparison across datasets by accounting for, rather than ignoring, the differences.

Materials & Methods

Two sets of experiments were conducted for this study (Table 1). In the first, the volume of water filtered and filter pore size varied while the preservation and extraction methods remained constant. This experiment also used homogenized source water to reduce true biological variability in replicate samples and specifically measure the effect of the difference in volume filtered and filter pore size. The second set of experiments held volume filtered and filter pore size constant while varying the type of preservation and the extraction method. Here, source water was not homogenized and true biological variability (i.e., bottle to bottle variation) is both observed and included. After DNA extraction, all extracts from both experiments followed the same procedures for total DNA quantification, assessment for inhibition, and quantification of target DNA via qPCR.

Table 1 Sampling details.

Details on the two marine eDNA field campaigns conducted in this study.

	Campaign 1	Campaign 2	
Date	September 2022	February 2023	
Environment	Net enclosure open to environment, ∼15 °C	Closed, recirculating, filtered pool, ∼20 °C	
Number of dolphins present at time of sampling	3	1	
Collection and filtration details	Filtration occurred 4 h after collection	Collection and filtration in situ (i.e., 0 h lag)	
Homogenization	Yes	No	
Volume filtered (L)	1, 3	3	
Pore size of filter (μ m)	1, 5	5	
Preservation method	Longmire’s buffer	Longmire’s buffer, −80 °C, RNAShield, Desiccation	
Extraction method	Phenol-chloroform-isoamyl (PCI)	PCI, Qiagen Blood and Tissue, Zymo Miniprep	

Field sampling, filtration, preservation, extraction

Campaign 1: pore size and volume comparison

Water samples were collected in Hood Canal near Bangor, Washington in September 2022 from a netted enclosure containing a small, managed population of Atlantic bottlenose dolphins (Tursiops truncatus). A total of 70 L of water was collected on site in large, clean carboys. Water was transported from the sampling site to the NOAA Northwest Fisheries Science Center (NWFSC) for processing, with an elapsed time of approximately 3 h.

At the lab, carboys were well mixed to homogenize the source water before splitting it into individual samples. Samples were assigned to a volume filtered treatment (1 L or 3 L) and a pore size treatment (1 µm or 5 µm), for a total of four treatments, each with three biological replicates (i.e., filters). All filter membranes were 47 mm diameter mixed cellulose ester (MCE) and water was filtered using a vacuum manifold and sterile, single use filter funnels. After filtration, filters were transferred with sterile forceps and to 5 ml tubes containing 2 ml of Longmire’s buffer (Renshaw et al., 2015) and stored at room temperature for 2 months until DNA extraction (Wegleitner et al., 2015). All filters were extracted using a phase lock protocol for phenol-chlorofom-isoamyl DNA purification (see Ramón-Laca, Wells & Park (2021) for detailed protocol). Total DNA was quantified via a Qubit fluorometer.

Campaign 2: preservation and extraction comparison

For the samples where preservation and extraction varied, water was collected in February 2023 from a closed pool with recirculating water and a filtration system with one dolphin inhabiting the pool at the time of sampling. Water samples were filtered on site in situ using a Smith Root Citizen Science sampler. For each water sample, 3 L of water was filtered onto a 5 µm pore size 47 mm diameter MCE self-preserving Smith Root filter (Thomas et al., 2019). A total of 45 filters were collected over approximately 1 h. The self-preserving filters were transported back to NOAA NWFSC within approximately 3 h, where samples were randomized across the time of collection to various preservation treatments (n = 4 total; Longmire’s buffer, −80 °C freezer, desiccation via self-preserving at room temperature, and Zymo DNA/RNA Shield) and extraction method (n = 3 total; phenol-chloroform-isoamyl alcohol, Qiagen Blood and Tissue Kit, Zymo Mini Prep Kit). Kits were scaled up accordingly to process 2 mL of preservative. Two combinations of preservation and extraction were not possible due to sampling limitations (Longmire’s buffer with Qiagen Blood and Tissue and Longmire’s buffer with Zymo Mini Prep Kit), resulting in not a full factorial design but 10 combinations of the two treatments. We were unable to complete the Longmire’s buffer with the two kits as we added the Longmire’s with PCI at the last minute to be able to link Campaign 2 to Campaign 1 by using the same preservation and extraction methods, but we did not collect enough additional samples to preserve in Longmire’s for the other two extraction methods. All samples were preserved for 3 months before being extracted. Total DNA of all extracts were quantified via a Qubit fluorometer.

Inhibition testing and target DNA quantification

All samples were assessed for inhibition by using an internal positive control (IPC) assay that was multiplexed with the target DNA assay by utilizing two different reporters. The IPC assay used was TaqMan Exogenous Internal Positive Control Reagents (EXO-IPC) (Applied Biosystems, Waltham, MA, USA). The IPC was included in all environmental samples and also in no template controls (NTC). Samples were deemed inhibited if the difference in mean Ct value of the IPC measured in the sample and the mean Ct value of the IPC measured in the NTC was greater than 0.5. Inhibited samples were diluted and re-run until the delta Ct was less than 0.5. The maximum dilution needed to alleviate inhibition was 1:100.

Target DNA (T. truncatus) was quantified using a regionally-specific assay (Brasseale et al., 2025) targeting an 80 base pair fragment of the cytochrome B gene region. The assay can also amplify species in the genera Stenella and Delphinus, however those are extremely rare or absent in Hood Canal, therefore we deem the assay as specific to T. truncatus in this particular region, but if it were used elsewhere where Stenella or Delphinus occur, multiple species may amplify. Standard curves were conducted using a synthetic gBlock (IDT) from 100,000 copies/µL to 1 copy/µL in a ten-fold dilution series. All samples were run in triplicate and each plate contained three no template controls (NTCs). The forward primer sequence was 5′-TTATTCTTCCATTCATCATCAC-3′, the reverse primer sequence was 5′-GTGGGGTTGTTGGATCCTGT-3′, and the probe sequence was JUN-GAATAGTAGGTGAACGGCTGCCA-QSY. Each reaction contained, per 10 µL reaction: 5 µL of TaqMan Gene Expression Master Mix (Applied Biosystems), 0.4 µL of 10 µM forward primer, 0.4 µL of 10 µM reverse primer, 0.2 µL of 10 µM probe, 2 µL of DNase/RNase free water and 2 µL of DNA. Reactions were with an initial denaturation of 95 °C for 10 min, followed by 45 cycles of denaturation (94 °C for 15 s) and annealing/extension (60 °C for 1 min). Final concentrations in field samples were corrected according to the dilution factor if dilution was required to alleviate inhibition. All NTC showed no amplification. The assay reliably detected the gBlock at concentrations of 10 copies/µL and stochastically detected the 1 copy/µL concentration.

Optimizing the target-to-total DNA ratio

Our other metric of interest in addition to absolute target DNA concentrations is the ratio of target DNA (as measured by qPCR) to total DNA (as measured by Qubit). The units of target DNA are copies per volume whereas total DNA is mass per volume. We converted the total DNA from mass to copy number by using the length of the fragment, the average mass of each base pair, and Avogadro’s number (Eq. 1). This gives us the concentration in copy/µL if all the total DNA in the extract were 80 base pair fragments (i.e., the length of the target DNA amplified by the T. tursiops assay). Rather than using the exact molecular weight of the target DNA, we use an average molecular mass of a DNA base pair (618 g/mole) given that we know that all genomic DNA is not target DNA. This allows us to make a ratio by having both quantities in the same units, using: (1) CtotalDNAcopies/μL=CtotalDNAng/μL ∗6.022 ∗1023copy/molMWg/mol ∗109ng/g.

Then the ratio of target to total DNA is calculated simply by: (2) Ratio=CtargetDNAcopy/μLCtotalDNAcopy/μL.

Linear models to compare different methodological choices

Here, we present two linear models to quantify how different methodological choices (volume filtered, filter pore size, preservation method, extraction method) impact target DNA quantification. Given that for each field campaign, the water sampled all comes from a common pool, we can attribute the differences in target quantification to (1) sampling variability, and (2) the differences in methodological choices in collection and processing. Because we did not have a full factorial sampling design over our two field campaigns, we developed two closely related models: one predicting concentration as a function of volume filtered and pore size, and the other predicting concentration as a function of preservation and extraction methods.

In both models, (log) observations of concentrations of target DNA are treated hierarchically, with observations from technical replicate i, bottle j, treatment k, and campaign m drawn from a nested series of normal distributions: yijkm∼Nμjkm,σ

μjkm∼Nθkm,τ

θkm∼Nϕm,ν

where μjkm is the mean of each biological replicate (i.e., bottle-level mean) and σ is the standard deviation among technical replicates within a biological replicate. Bottle-level means are in turn treated as samples from a treatment-level distribution of mean θkm and standard deviation τ. Finally, all treatment-level means are treated as draws from a campaign-level (overall) distribution of mean ϕm and standard deviation ν. The model reflects the fact that the samples are nested, with samples from each campaign collected from a common pool of water, but each treatment (here, pore size and volume) has a set of biological replicates, and those biological replicates are subsamples of the common pool and technical replicates are subsamples of each biological replicate.

We first model the effects of volume filtered and pore size filter across two field campaigns as: (3) θkm=βX

where β is a vector of regression coefficients having the same length as θ, and X is the design matrix mapping different combinations of volume filtered and pore size (i.e., treatments) to the data.

The second model is identical, but for treatments of different preservation and extraction methods, and having only a single campaign, such that it requires only the first two hierarchical levels (i.e., there is no campaign-level mean to separate the two pools of water).

The priors for both models were the same and were as follows: μjkm∼N0,10

σ∼gamma1,1

τ∼gamma1,1

β∼N0,5.

The models were both implemented in RStan.

Results

Total DNA, target DNA, ratio of target to total DNA

For the samples collected in Campaign 1, we compared volume filtered and filter pore size (Table 2, Fig. 1). Given the same volume of water filtered, the 1 µm filters had higher total and target DNA as compared to the 5 µm filters. Given the same pore size, the 1 L filters had less total and target DNA than the 3 L filters. However, the 1 µm filters with 3x volume filtered had more than 3x total DNA (340% of mean value) and more than 3x target DNA (434% of mean value). The 5 µm filters showed a similar pattern with the 3x volume samples having more than 3x total DNA (325% of mean value) but showed much more than 3x target DNA (514% of mean value). When converting these to ratios of target to total DNA, 1 µm filters were lower than 5 µm filters and the 1 L samples were lower than the 3 L samples, resulting in the 5 µm and 3 L filter having the highest ratio of target to total DNA.

Table 2 Volume and pore size experiment results.

For two volumes filtered and two pore sizes, the mean and standard deviation of total DNA (ng/uL) and target DNA (copies/uL) recovered, and the ratio of mean target to total DNA.

Volume filtered (L)	Filter pore size (μ m)	Mean total DNA (ng/μ L)	Standard deviation total DNA	Mean target DNA (copies/μ L)	Standard deviation target DNA	Ratio target: total DNA	
1	1	51.4	3.93	7,172	1,030	1.14e−07	
1	5	39.9	6.08	5,908	879	1.22e−07	
3	1	175	20.7	31,191	2,294	1.48e−07	
3	5	127	11.6	30,385	1,540	1.99e−07	

Figure 1 Total DNA, target DNA, and ratio from volume and pore size experiment (Campaign 1).

(A) Total DNA recovery (ng/uL) as a function of volume of water filtered and filter pore size. (B) Target DNA (copies/uL) as a function of volume of water filtered and filter pore size. (C) The ratio of target to total DNA as a function of volume of water filtered and filter pore size.

Here, the 1 µm filters captured more total DNA than the 5 µm filters did, presumably across a range of particle sizes; we assume that the material captured on a 5 µm filter is a subset of the material on a 1 µm filter, with the larger pore size selecting for larger particles and not capturing particles <5 µm (at least initially; at some point as the filter clogs it will have an effective pore size smaller than 5 µm and capture smaller particles). To the extent that our target eDNA fragment—from vertebrate mtDNA—is more likely to occur in larger particles (due to the size of mammalian cells, etc.), we expect and observe a larger target:total ratio with 5 µm filters (Power et al., 2023). In terms of inhibition, all 1 L samples required 1:20 dilutions except two technical replicates requiring 1:100 dilutions, whereas the 3 L samples required dilution from 1:10 to 1:40 (Figs. S1 and S2). However, there was no significant effect of either volume filtered or pore size on the dilution factor required to alleviate inhibition.

For the second campaign comparing preservation and extraction methods, the total DNA varied across methods, with PCI extractions having consistently higher yields than either kit across preservation methods (Fig. 2). However, target DNA was similar across preservation and extraction methods. The samples preserved via −80 °C had the highest target DNA recovery, but the resulting ratio of target to total DNA exhibits the opposite trend of the total DNA yield, where PCI has the lowest ratio, then the Qiagen kit, and finally the Zymo kit with the highest ratio (i.e., the most desirable for a particular targeted assay). For all samples extracted with the Zymo kit, the Zymo DNA/RNA Shield had the highest ratio of the three preservatives tested. We found that the samples extracted via PCI had the highest total DNA concentrations however had the most samples that were not inhibited at 1:1 compared to Zymo and Qiagen (Figs. S3 and S4).

Figure 2 Total DNA, target DNA, and ratio from preservation and extraction experiment (Campaign 2).

(A) Total DNA recovery (ng/uL) as a function of preservation method and extraction method. (B) Target DNA (copies/uL) as a function of preservation method and extraction method. (C) The ratio of target to total DNA as a function preservation method and extraction method.

Biological variability with homogenized and non-homogenized samples

In the first campaign, a large volume of water was collected and homogenized before splitting into the different treatments of pore size and volume filtered. In the second campaign, individual water samples were grabbed from the source (well mixed, relatively small pool) but not homogenized before filtering. In both experiments, the coefficient of variation between technical replicates (Fig. 3, X markers) was less than the coefficient of variation between biological replicates (Fig. 3, colored circles), which was less than the coefficient of variation between treatments (Fig. 3, dashed lines; except in two sets of biological replicates). This trend demonstrates that it is important to have replication to be able to separate technical and biological variability from the treatment effect (here, volume/pore size and preservation/extraction).

Figure 3 Coefficient of variation (CV) across campaigns.

(A) CV of target DNA as a function of volume of water filtered, colored by pore size filter, between technical replicates (Xs) and biological replicates (circles) in homogenized water from Campaign 1. (B) CV as a function of preservation method, colored by extraction method, from non-homogenized water from Campaign 2. Dashed lines show the CV from all samples across all treatments. In (A), the dashed line shows all treatments but the dotted line shows the coefficient of variation if the concentrations are adjusted for the volume filtered in the experiment.

For the volume and pore size experiment, we also calculated the coefficient of variation assuming a linear increase in concentration with volume filtered (Fig. 3, Panel A, dotted line). The coefficient of variation across treatments in the homogenized water in Campaign 1 (pore size and volume) and adjusting for the volume filtered was 0.24, compared to a coefficient of variation of 0.35 across treatments in Campaign 2. For the homogenized samples, the mean coefficient of variation between biological replicates was 0.1, whereas the mean coefficient of variation between biological replicates in the non-homogenized samples was 0.2 (Fig. 3). The observed biological variability did not seem to be related to the methodological choices (Fig. 3). Given how close the biological variability was to the variability across treatments especially in the preservation and extraction experiment, the linear models are particularly helpful to distinguish sources of variability.

Models to combine samples processed with different methods

Our two sets of experiments provided an opportunity to use two models to combine samples with different methodological choices from the same common reality (e.g., different pore size and volume filtered in the first experiment and different preservation and extraction methods in the second experiment). Because in a single campaign, the samples came from the same source water (and in the case of the first experiment, the source water was homogenized), we can assess how the different methodological choices affect differences in target DNA recovery through the use of linear models.

Pore size and volume sampled

The first linear model was used to investigate the effects of different filter pore sizes and volumes on target DNA concentrations (Table 3, Fig. 4, Fig. S5). Each campaign has its own intercept, reflecting the two different concentrations of target DNA in the underlying pools of water sampled for each campaign. We quantified different treatment effects relative to the base case of filtering 1 L through a filter of 1 µm pore size; the coefficients for the effects of different treatments reflect departures from this baseline method. As with any similar analysis of variance (ANOVA), the choice of reference condition is arbitrary and does not affect the conclusions.

We find no meaningful marginal effect of changing from 1 µm to 5 µm pore size, after controlling for volume filtered (Table 3), but do see a greater-than-expected increase in target concentration by increasing sampling effort from 1 L to 3 L of water filtered. Rather than a linear increase with volume filtered (i.e., a parameter value of 1), we find an estimated scaling factor of 1.43 (95% posterior CI [1.14–1.71]).

Preservation and extraction methods

Here, the source water was the same for all samples so there is a single intercept, which represents preservation by −80 °C and extraction by PCI (i.e., the base case for treatment type); again, the coefficients for treatment effects indicate deviation from those methodological choices (Table 4, Fig. 5, Fig. S6). We find no meaningful effects of preservation or extraction method—including interactions among a subset of these—on our target eDNA concentration, with 1 exception: desiccation as a preservation method yielded systematically less target eDNA than other methods (mean posterior coefficient estimate = −0.894, 95% posterior CI [−1.37 to −0.429]). Desiccation retains approximately 40% of target eDNA relative to what would be retained by preserving filters at −80 °C immediately after filtering. Across a broad range of methodological choices, our results suggest rougly even performance in recovery of target DNA with only a few departures.

Table 3 Model parameters for volume and pore size experiment.

Coefficients in bold are meaningfully different than zero.

Parameter	Mean estimate	Standard deviation	2.5% CI	97.5% CI	
Campaign 1	8.81	0.134	8.55	9.09	
Campaign 2	11.2	0.271	10.6	11.7	
Pore size (5 μ m)	−0.0937	0.157	−0.409	0.214	
Log volume	1.43	0.140	1.14	1.71	

Figure 4 Volume and pore size linear model results.

The model uses technical replicates (grey x’s) to generate biological replicate, or bottle, means (blue circles), and bottle means are then fed into treatment means (yellow circles). Error bars show 2.5% and 97.5% confidence intervals.

Table 4 Model parameters for preservation and extraction experiment.

Coefficients in bold are meaningfully different than zero.

Parameter	Mean estimate	Standard deviation	2.5% CI	97.5% CI	
(Intercept)	12.8	0.166	12.5	13.1	
Desiccation	−0.894	0.239	−1.37	−0.429	
Shield	−.0330	0.236	−0.807	0.133	
Longmires	−0.130	0.234	−0.593	0.320	
Qiagen	0.195	0.238	−0.281	0.660	
Zymo	0.105	0.236	−0.365	0.555	
Desiccation/Qiagen	0.167	0.339	−0.492	0.840	
Shield/Qiagen	−0.00194	0.337	−0.673	0.672	
Desiccation/Zymo	0.400	0.333	−0.261	1.05	
Shield/Zymo	0.124	0.337	−0.523	0.821	

Figure 5 Preservation and extraction linear model results.

The model uses technical replicates (grey x’s) to generate biological replicate, or bottle, means (blue circles), and bottle means are then fed into treatment means (yellow circles). Error bars show 2.5% and 97.5% confidence intervals.

Harmonizing samples across different protocols

Therefore, for a given sample that was preserved or extracted one way, we can then calculate an adjusted concentration to reflect what the concentration likely would have been given a different preservation or extraction method. For example, if we wanted to combine the data from Campaign 1 with the data from Campaign 2, we would take the intercept for Campaign 1 (8.81), add −0.0937 for the 5 µm filter pore size and add 1.43*log(3) for the volume filtered and have a log DNA concentration of 10.29 copies/µL, which was preserved in Longmire’s buffer and extracted via PCI. If we wanted to make this value comparable to the water sampled in Campaign 2 (which was 3 L filtered on 5 µm filter with the intercept as −80 °C and PCI), we would take the intercept 12.8 and add −0.130 for the switch of preservation from −80 °C to Longmire’s buffer to get a log DNA concentration of 12.67 copies/µL. Moving those out of log space, we can compare 29,436 copies/µL from Campaign 1 to 318,062 copies/µL in Campaign 2. Now we can quantitatively compare these concentrations because we have modeled the effects of different treatments explicitly. We can do this for any observational data where we have the combination of methods in our matrix and can translate.

Discussion

More (total) DNA is not always better

The natural inclination is to maximize capture of all DNA in order to capture more target DNA, either by using a smaller pore size filter or using a different extraction protocol. We found here that, as expected, both target and total DNA increased with both increased sample volume and with smaller pore size filters (Capo et al., 2020). However, the increase in target DNA recovery was small compared to the increase in total DNA recovery. In other words, the target was a smaller percentage of total DNA (in the case of 1 µm filters, total DNA increased by 340% from 1 to 3 L while target DNA increased by 434% and in the case of 5 µm filters, total DNA increased by 318% from 1 to 3 L while target DNA increased by 514%). By having a rare target in a larger pool of off-target DNA, other issues can arise associated with the concentration of various inhibitors. In both experiments, we demonstrate that the optimal methods for maximizing total DNA capture do not maximize the target DNA nor the ratio of target to total DNA. Here, we did not find that samples with more total DNA had more inhibition, but in cases where the target DNA is very rare, the maximization of target/total DNA should be carefully considered. For example, if the absolute target DNA concentrations is 9 copies/L and a 1 L sample (9 copies total) is not inhibited but a 3 L sample (27 copies total) requires a 1:100 dilution, the target DNA would be diluted (perhaps beyond the limit of detection) in the inhibition treatment.

Though inhibition was not at a threshold where the rarity of the target DNA in a larger pool of total DNA was an issue, there are remarkable patterns in particularly the extraction method (Fig. 2C) demonstrating the increasing ratio of target to total DNA recovery. As noted above, we encourage researchers to consider not only the absolute recovery of either target or total DNA, but to consider when target is particularly rare and inhibition is likely to choose a methodology focusing on maximizing the ratio rather than absolute concentration.

The same concepts apply with the preservation and extraction experiment with different methodological choices resulting in different total DNA capture, target DNA capture, and ratio of target to total DNA capture (Spens et al., 2017; Deiner et al., 2015; Deiner et al., 2018; Djurhuus et al., 2017). Longmire’s buffer with PCI extraction yielded the highest total DNA concentration while −80 °C preservation with Qiagen extract yielded the highest target DNA concentration, however Shield preservation with Zymo extraction yielded the highest ratio of target to total DNA. As with any ecological sampling, different methods can and do result in different reflections of the environment being sampled. However, a quantitative framework accounting for such methodological variability—such as the one we present here—offers a simple means of combining information across protocols.

Some hypotheses of the differences in total, target, and ratio of total to target DNA across different pore sizes, preservation, and extraction methods include consideration of the mechanisms of filtration and the combinations and compatibility of preservatives and extraction kits (Capo et al., 2020). For the filter pore size, smaller pore sizes (here, 1 µm) will capture more bacteria which will contribute to the total DNA yield, but as in this study targeting dolphin DNA, are non-target (Power et al., 2023). A larger pore size filter (here, 5 µm) will result in less capture of smaller organisms such as bacteria and will leave room for capturing more of the desired larger animal cells before the filter clogs or the designated volume has been filtered. It should be noted that this study was conducted in relatively shallow, near-shore, nutrient rich water. Deep sea or oligotrophic waters might concentrate less off-target DNA and the ratios found here are not necessarily portable to different environments.

We note that it would have been informative to have a wider range of absolute concentrations of target DNA and a wider range of ratios of target to total DNA to explore these methodological choices more. In particular, it would be informative to have lower target DNA concentrations where the trade-offs between inhibition treatments (namely, dilution) and target recovery are empirically demonstrated. We also acknowledge that a wider range of pore sizes and volumes filtered for the first campaign would have been particularly informative to confirm the trends observed. Future studies could expand to even smaller pore size filters (e.g., 0.22 or 0.45 µm) and even larger pore size filters (e.g., 10 µm). Finally, we acknowledge the limitation of using a single species target and quantification via qPCR, which limits the applicability to other single-species targets with quantification via qPCR or to metabarcoding studies. However, we expect that the mechanisms we discuss here are broadly applicable and generalizable to macro-organisms. For single-species targets and quantification via qPCR for micro-organisms, in particular, the pore size of the filter should be smaller for optimal capture. For metabarcoding, we expect the findings here to be broadly generalizable to primers targeting macroorganisms as well, though we hypothesize that there could be a larger effect in lower concentration samples given the compositional nature of metabarcoding data.

Even when collecting water while looking at a dolphin, we are still looking for a needle in a haystack

We discussed maximizing the target to total DNA ratio to minimize the rarity of the target relative to non-target DNA, however it is worth noting the absolute values of that ratio, especially given how sampling was conducted. In both experiments, dolphins were present while the water was being collected. The absolute concentration of dolphin DNA was high, as expected, and similar order of magnitude concentrations found in other studies where sampling is conducted in close proximity to the target organism (Brasseale et al., 2025) and lower than others with many more individuals present (e.g., Capo et al., 2020; Eichmiller, Miller & Sorensen, 2016). To generate the ratio of target to total DNA, the total DNA concentration had to be converted from units of mass per volume as measured by the Qubit fluorescence reader to copies per volume to match the units of the target DNA concentration as measured by qPCR. This conversion requires a fragment length, which here we use the length of the fragment targeted by the qPCR assay. Therefore, the denominator used in the ratio can be thought of as the total number of fragments that could possibly be target (here, dolphin).

These percentages are very low. Even when sampling water while looking at the species of interest and using methods intended to maximize the target to total ratio (here, larger pore size filter and larger volume of water), the target is just 0.00001% of the total DNA. This corresponds relatively well with a study that used shotgun sequencing and found fish to be 0.00004% of the total reads from a 1 L water sample taken from a reef in Australia and filtered on a 0.2 µm nylon filter, frozen at −20 °C, and extracted using the Qiagen Blood and Tissue Kit (Stat et al., 2017). It is worth noting that water samples contain genetic material from many, many species that we are not interested in, and this is important to bear in mind within the context of very rare targets and the possibility of false negatives.

Here, our molecular assay reliably detected 10 copies/µL of extract. Most extraction protocols (including the one we followed) elute extracts in 100 µL. If 1 L of water was filtered, this becomes 1,000 copies/L filtered, whereas if 3 L of water was filtered, this becomes 333 copies/L, so by filtering larger volumes, more dilute DNA can be detected. Furthermore, a single cell can have 100s to 10,000s of mitochondria, and in humans, each mitochondrion has 2–10 copies of mtDNA, resulting in a range from 100s to 100,000s of copies of mtDNA per cell (Zhang et al., 2015; Castellani et al., 2020; Rath et al., 2024). A single cell is therefore easily detectable given the observed limit of detections.

There are many choices and most are just fine

Though the total DNA yield varied with preservative and extraction choice, the amount of target DNA recovered was similar across methodological choices. Some preservative and extraction protocol combinations are less than ideal (i.e., desiccation with PCI), but most methodological combinations perform similarly. In contract, the −80 °C preservation had the highest target DNA recovery, but can provide to be logistically challenging or impossible in the field. Therefore, practicality of methodological choices must also be considered. Additionally, when selecting a combination of preservation and extraction methods, it is important to keep in mind the mechanisms of the different preservatives. For example, Longmire’s buffer and DNA/RNA Shield are both lysis buffers, meaning that the DNA is actually preserved in the buffer and removed from the filter while the filter is submerged in the buffer. On the other hand, the self-preserving filters and storing filters in the −80 °C both work by desiccation and therefore the DNA is still on the filter when starting the extraction.

Accordingly, as long as the extraction method is compatible with whether the DNA is still on the filter or in the buffer, there should not be large differences in target DNA yield. Again, the total DNA yield will differ based on extraction protocol (e.g., PCI will recover more total DNA), but the target DNA seems relatively robust to different preservation and extraction methods assuming compatibility between the two. Note, there were differences in target DNA recovery with preservation method with −80 °C having higher target DNA concentrations, but logistics also must be considered. Access to reagents and infrastructure like freezers may vary and logistical constraints may impact the decisions for preservation and extraction methods.

Responsibly combining data from different methods

Particularly relevant for time series data or combining data from different projects, it is important to keep methods consistent. However, there are many reasons why one might want to combine data generated from different protocols. Here, we demonstrate the use of simple linear models to “correct” for different protocols and make data comparable. It is important to have a calibration experiment where all possible combinations of protocols are sampled from a common reality in order to make the corrections. However, once that has been done, this allows for extrapolating any scenario from the linear model for unknown samples. Importantly, any set of samples can be translated to the equivalent concentration of a different methodological choice. This is essential to responsibly combine quantitative data collected via different methods, thereby facilitating the generation of larger sample sizes and larger spatial and temporal coverage for exploring broad-scale hypotheses. Future work could look at doing something similar with different species-specific assays for the same target species and determining how portable these parameter estimates are in relation to different assays and other water samples, especially considering other environmental factors like turbidity, salinity, or other parameters that might affect portability.

Conclusions

Many publications exist comparing results from different methodological choices in eDNA protocols. However, here we approached this methodological comparison with a very specific goal of defining the metric to maximize, the ratio of target DNA to total DNA for a single species quantitative assay. We find that while smaller pore size filters collect more total DNA, the target DNA is similar and therefore a larger pore size filter with larger volumes of water filtered maximizes the target to total DNA ratio recovered. We also find that while different preservatives and extraction methods vary, the variance tends to be reflected by larger changes in total DNA yield rather than target DNA yield. Accordingly, we found that the extraction method with the highest target to total DNA ratio was from the commercially available kits rather than PCI. Finally, we introduce simple linear models to correct data sourced from samples with varying protocols, allowing researchers to utilize information from varying protocols in a responsible manner.

Supplemental Information

Supplemental Information 1 Inhibition testing results from Campaign 1 (pore size / volume filtered)

The y axis shows the difference in Ct value from the environmental sample versus the no template control of the spiked internal positive control (IPC). The x axis shows the total DNA concentration of the sample as measured by Qubit. Colors correspond to the pore size of the filter (um) and shapes correspond to the volume of water filtered (L). Dashed lines represent the threshold at which samples were deemed inhibited (0.5 Ct difference). Facets indicate the dilution factor.

Supplemental Information 2 Inhibition testing results from Campaign 1 (pore size / volume filtered); same data as in Figure S1

The y axis shows the difference in Ct value from the environmental sample versus the no template control of the spiked internal positive control (IPC). The x axis shows each unique sample. The facets correspond to the pore size of the filter (um) and shapres correspond to the volume of water filtered (L). The colors show the dilution factor.

Supplemental Information 3 Inhibition testing results from Campaign 2 (preservation and extraction method)

The y axis shows the difference in Ct value from the environmental sample versus the no template control of the spiked internal positive control (IPC). The x axis shows the total DNA concentration of the sample as measured by Qubit. Colors correspond to the extraction method and shapes correspond to the preservation method. Dashed lines represent the threshold at which samples were deemed inhibited (0.5 Ct difference). Facets indicate the dilution factor.

Supplemental Information 4 Inhibition testing results from Campaign 2 (preservation and extraction method); same data as in Figure S3

The y axis shows the difference in Ct value from the environmental sample versus the no template control of the spiked internal positive control (IPC). The x axis shows each unique sample. The facets correspond to the preservation and extraction method. The colors show the dilution factor.

Supplemental Information 5 MIQE Checklist

Supplemental Information 6 Volume and Pore Size Linear Model Results

Modeled estimates versus the observed mean of technical and biological replicates. Error bars show 2.5% and 97.5% confidence intervals. Colors correspond to the pore size of the filter (um) and shapes correspond to the volume of water filtered (L).

Supplemental Information 7 Preservation and Extraction Model Results

Modeled estimates versus the observed mean of technical and biological replicates. Error bars show 2.5% and 97.5% confidence intervals. Colors correspond to the extraction method and shapes correspond to the preservation method.

The authors are grateful to US Naval Base Kitsap-Bangor for logistic support. This study was supported through collaborative research efforts with the US Navy’s Marine Mammal Program. Any use of trade, firm, or product names is for descriptive purposes only and does not imply endorsement by the US Government.

Additional Information and Declarations

Competing Interests

Author Contributions

Data Availability

The authors declare there are no competing interests.

Elizabeth Andruszkiewicz Allan conceived and designed the experiments, performed the experiments, analyzed the data, prepared figures and/or tables, authored or reviewed drafts of the article, and approved the final draft.

Megan R. Shaffer conceived and designed the experiments, performed the experiments, analyzed the data, prepared figures and/or tables, authored or reviewed drafts of the article, and approved the final draft.

Ryan P. Kelly conceived and designed the experiments, authored or reviewed drafts of the article, and approved the final draft.

Kim Parsons conceived and designed the experiments, authored or reviewed drafts of the article, and approved the final draft.

The following information was supplied regarding data availability:

The data and code are available at Github and Zenodo:

- https://github.com/eandrusz/MURI_Mod1_Needle_Haystack

- Elizabeth Andruszkiewicz Allan, & Ryan Kelly. (2025). eandrusz/MURI_Mod1_Needle _Haystack: Optimizing target-to-total DNA ratio in eDNA studies (v1.1.1). Zenodo. https://doi.org/10.5281/zenodo.16858968.

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
