# Peer review of "Optimizing target-to-total DNA ratio in eDNA studies: effects of sampling, preservation, and extraction methods on single-species detection"

_PeerJ, doi:10.7717/peerj.20127_

## Round 0.1 · original submission · Major Revisions

· Academic Editor

Major Revisions

Dear authors,

Three experts in the field have read your manuscript and agree that your manuscript is interesting and contains important information for eDNA isolation. However, some of the issues and recommendations are highly relevant. For example, the number of samples, the distance to the dolphins, and more importantly, an explanation of the enrichment of target DNA dependent on the method of isolation, which is not clear to the reviewers or me. The manuscript is highly suitable for the PeerJ readership, so I kindly request that you address the reviewers' comments and return the paper for evaluation.

Thank you so much for choosing PeerJ.

Good luck with your research moving forward.

Best regards,
Bernardo

Reviewer 1 ·

Basic reporting

no comment

Experimental design

no comment

Validity of the findings

no comment

Additional comments

In the manuscript :Not your average environmental DNA methods paper: Evaluating the effects of sampling, preservation, and extraction methods and target species yields, the authors show how the quantity and quality of environmental DNA is affected according to parameters such as the amount of sample taken from the environment (liquid), the size of the filter used to collect the sample, the method of preservation and extraction of the eDNA. they take as an example the detection of DNA from a dolphin species.

The paper is interesting because it explicitly exposes some of the variables that are important to evaluate the presence of a specific environmental DNA.


The authors show through several tests which is the best condition to extract eDNA through the ratio of target DNA to total DNA. In all of the conditions presented a positive result is obtained, i.e. the presence of target DNA (dolphin DNA), although the quantification is important, I believe that the important is to know the presence of the target DNA, whether it is a lot or a little is not relevant, since in any case the conclusion is the presence of the animal or not. So in this case knowing the best way to take and process the sample is not relevant. In my case I would choose the cheapest and easiest method to handle and not necessarily the one that gives me the best ratio of white DNA to total DNA.

It would have been interesting to have a case where this ratio is important to detect whether or not there is another target DNA.

I don't understand why getting more total DNA, which includes the so-called off target DNA, which is actually all the DNA except for the target DNA, is detrimental to getting a positive result. i.e. why does having more DNA make it more difficult to detect the target DNA?


Why the filtered samples (the filters) in campaign 1 or 2 were stored at room temperature for 2 or 3 months, this is not detrimental to the amount of DNA obtained, is it not better to store these samples in these 2 or 3 months at 4°C at least? in an article cited by the authors the eDNA is stored for two weeks, but in the case of the present study it is stored 8 times longer at room temperature.


As for the models; the mathematics is poorly explained considering that the readership of this journal is of general scientific interest. In addition, I was unable to obtain the supplementary figures 5 and 6 cited in the manuscript (338 and 356) which are used to explain the models described.


Looking at Figure 1 and 2 panel C, where the ratio in any case is of the same order of magnitude for the different assays, so you don't really get more eDNA that has consequences on the identification of the target DNA.


I think experiment 1 would have been more complete and if the non-homogenized water condition had been included to see if in the same experiment, without a model, there is an effect.

In the comparisons between campaigns 1 and 2 with the model used, do you take into account that in one case the sample was homogenized and in the other it was not? is this relevant? or that the samples were taken in different environments (Table 1)?

In general terms the discussion lacks references where the data obtained here are compared with other works, in fact there is only one reference in the whole discussion, the results could be compared with the results obtained by the same authors in: https://doi.org/10.1029/2024JC021643


minor corrections:
update the reference: Brasseale, E., N. G. Adams, E. Allan, E. K. Jacobson, R. P. Kelly, O. R. Liu, S. Moore, M. Shaffer, J. Xiong, and K. Parsons. (n.d.). Marine eDNA production and loss mechanisms.

Reviewer 2 ·

Basic reporting

The paper discusses the methods and their efficiencies for collecting and purifying total DNA and mitochondrial DNA (mtDNA) from water collected in dolphin-hosting ponds. It focuses its observations on the capture of total DNA (measured spectrophotometrically) versus mtDNA from which cytochrome B was amplified.
The English used is adequate; they use appropriate references for their observations, and their tables, statistics, and analyses are well presented. However, the observations are particular, of limited interest to people with similar problems, so their hypotheses are also limited in terms of their generalizable academic value.

Experimental design

Methodologically, this leaves several unanswered questions, such as whether the water collected on both occasions is comparable in terms of pH, solids in solution, temperature, and the presence of nitrates and nitrites, among other factors that could bias the observations. It is also unclear how far the dolphins were from the water sampling site, and whether these distances were equivalent in both cases. It is also unclear what cell types are believed to be the DNA donors, and whether this could be demonstrated. Researchers in this area are undoubtedly interested in defining optimal sampling methods for working with DNA in water; however, these remain specific observations of interest to a limited audience.
One of the points that requires further development is the explanation of the results, specifically the case where the highest amount of total DNA collected is not accompanied by specific DNA (in this case, mtDNA). Proposed explanations must be tested, not just suggested.

Validity of the findings

Water sampling methods for DNA analysis are continuously being evaluated, especially for tracking etiological agents in urban wastewater and for identifying epidemic outbreaks. Collecting seawater is also a crucial aspect of controlling pollutants by monitoring organisms affected by such contamination. Thus, this is an ongoing and often debated issue, which reduces the novelty of this work. However, its results are thoroughly analyzed, although the proposed explanations necessitate experimental work.

Additional comments

No coments

·

Basic reporting

The manuscript by Andruszkiewicz Allan et al. provides a robust evaluation of sampling, preservation, and extraction methods for environmental DNA (eDNA) detection, focusing on optimizing the target-to-total DNA ratio for single-species detection of the Atlantic bottlenose dolphin using qPCR. Unlike typical eDNA methods papers, it introduces a novel statistical framework for harmonizing data across different protocols, offering a valuable tool for integrating datasets in eDNA research. The manuscript is well-written, with clearly presented methods, rigorous data analysis, and well-supported conclusions. However, I have a few minor concerns and suggestions to further strengthen the paper.

Experimental design

I have some suggestions, see the last section.

Validity of the findings

they are valid, but may be strengthen, see last section.

Additional comments

Minor concerns:

1) Title Revision for Clarity

While the title, “Not Your Average Environmental DNA Methods Paper: Evaluating the Effects of Sampling, Preservation, and Extraction Methods and Target Species Yields,” is engaging, it could better highlight the study’s key finding: the importance of optimizing the target-to-total DNA ratio rather than maximizing total DNA yield. I suggest revising the title to emphasize this metric, such as:
“Optimizing Target-to-Total DNA Ratio in eDNA Studies: Effects of Sampling, Preservation, and Extraction Methods on Single-Species Detection”

2) Acknowledgment of Non-Factorial Design

The study’s experimental design is not fully factorial, as some combinations (e.g., Longmire’s buffer with Qiagen Blood and Tissue or Zymo Mini Prep kits) were not tested. This limits the ability to compare all methodological combinations comprehensively. I recommend adding a brief discussion in the Methods or Discussion section to acknowledge this limitation and explain why certain combinations were excluded (e.g., logistical constraints). This would enhance transparency and guide readers in interpreting the results.

3) Scope Limitation to Single-Species qPCR

The study focuses on a single species (T. truncatus) and qPCR, which may limit its direct applicability to metabarcoding or other taxa. While the principles (e.g., optimizing target-to-total DNA ratio) are likely generalizable, this should be explicitly addressed. I suggest adding a paragraph in the Discussion section to note that further validation is needed for metabarcoding or multi-species assays and to discuss potential differences in applying these methods to other taxa (e.g., microbes vs. macroorganisms).

4) Expanding Pore Size and Volume Ranges

The conclusions regarding filter pore size (1 µm vs. 5 µm) and water volume (1 L vs. 3 L) are well-supported but based on a limited range of values. Testing additional pore sizes (e.g., 0.45 µm, 10 µm) and volumes (e.g., 0.5 L, 5 L) could strengthen the findings by confirming whether the observed trends (e.g., higher target-to-total DNA ratio with larger pore sizes and volumes) hold across a broader range. I suggest noting this as a limitation in the Discussion and recommending future studies to explore a wider range of parameters to enhance the robustness of the conclusions.

5) Clarification on Variation in Total vs. Target DNA Across Taxa

The Introduction states, “Other comparisons look at maximizing total DNA, assuming that more total DNA will yield more target DNA. However, this may not always be true and may vary with the target taxa (i.e., microbes vs. metazoans).” This claim could be clarified to explain why this variation occurs. For example, microbial DNA is often more abundant in environmental samples due to higher cell densities and smaller particle sizes, which smaller pore filters (e.g., 0.45 µm) capture more effectively. In contrast, metazoan DNA (e.g., from dolphins) is associated with larger cells or tissue fragments, which larger pore filters (e.g., 5 µm) capture more selectively, reducing non-target DNA.

6) Detection Limits and Influence of Target Organism Concentration


The manuscript does not explicitly address the detection limits of the qPCR assay or how varying concentrations of the target organism (dolphins) might affect the results. The study notes that target DNA was only ~0.00001% of total DNA, even with dolphins present during sampling, suggesting detection challenges for rare targets. Lower dolphin densities could reduce target DNA concentration below the assay’s limit of detection increasing false negatives, while higher densities could improve detection and reduce the sensitivity to methodological choices.

---

## Round 0.2 · accepted · Accept

· Academic Editor

Accept

Dear authors,

Based on the reports of the three experts in the field, your work has undergone thorough revision, and the manuscript is now suitable for publication. I thank the authors for choosing PeerJ for this interesting work. Thank you so much, and good luck with your research moving forward.

Best regards,
Bernardo

Reviewer 1 ·

Basic reporting

no comment

Experimental design

no comment

Validity of the findings

no comment

Additional comments

The authors responded extensively to the modifications suggested by the reviewers, which improves the understanding of the work.

Reviewer 2 ·

Basic reporting

No comment.

Experimental design

No comment

Validity of the findings

No comment

Additional comments

The clarification and expansion made by the authors in the introduction, in their second submission, improved the presented work.

·

Basic reporting

OK

Experimental design

OK

Validity of the findings

OK

Additional comments

Thanks for addressing all my comments!